# IoT Multi-Vector Cyberattack Detection Based on Machine Learning Algorithms: Traffic Features Analysis, Experiments, and Efficiency

Sergii Lysenko [1,*], Kira Bobrovnikova [1], Vyacheslav Kharchenko [2,*] 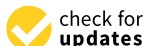 and Oleg Savenko [1]

1    Computer Engineering and Information Systems Department, Khmelnytskyi National University,
     29016 Khmelnytskyi, Ukraine; bobrovnikova.kira@gmail.com (K.B.); savenko_oleg_st@ukr.net (O.S.)
2    Department of Computer Systems, Networks and Cybersecurity, National Aerospace University "KhAI",
     61001 Kharkiv, Ukraine
*    Correspondence: sirogyk@ukr.net (S.L.); v.kharchenko@csn.khai.edu (V.K.); Tel.: +380-68-772-81-79 (S.L.);
     +380-67-915-19-89 (V.K.)

**Abstract:** Cybersecurity is a common Internet of Things security challenge. The lack of security in IoT devices has led to a great number of devices being compromised, with threats from both inside and outside the IoT infrastructure. Attacks on the IoT infrastructure result in device hacking, data theft, financial loss, instability, or even physical damage to devices. This requires the development of new approaches to ensure high-security levels in IoT infrastructure. To solve this problem, we propose a new approach for IoT cyberattack detection based on machine learning algorithms. The core of the method involves network traffic analyses that IoT devices generate during communication. The proposed approach deals with the set of network traffic features that may indicate the presence of cyberattacks in the IoT infrastructure and compromised IoT devices. Based on the obtained features for each IoT device, the feature vectors are formed. To conclude the possible attack presence, machine learning algorithms were employed. We assessed the complexity and time of machine learning algorithm implementation considering multi-vector cyberattacks on IoT infrastructure. Experiments were conducted to approve the method's efficiency. The results demonstrated that the network traffic feature-based approach allows the detection of multi-vector cyberattacks with high efficiency.

**Keywords:** Internet of Things; cybersecurity; cyber threats; malware detection; machine learning; network traffic

## 1. Introduction

### 1.1. Motivation

The Internet of Things is a concept that aggregates many technologies and physical objects—devices that exchange data and interact over the internet, as well as big data that generate these devices. Internet of Things devices have various purposes and complexities, from wearable things or technology to intelligent devices in smart homes and critical infrastructure. The Internet of Things was designed to make many areas of human life more comfortable and safer. However, the Internet of Things not only brings increased comfort but also new challenges and problems related to cybersecurity [1,2].

Security issues surrounding the Internet of Things infrastructure are determined by the specific features of an environment. One possible feature involved in building an IoT infrastructure is an IoT system of groups of identical or similar technical characteristic devices. If a specified device has a vulnerability, such homogeneity multiplies its impact [3–5].

Important issues include security issues with protocols used in the internet infrastructure, the use of unsafe network services, such as Telnet and SSH, and vulnerabilities in routers and open ports. With the ability to monitor and collect data on the IoT, even specialized compromised IoT devices with limited resources can be used to leverage critical



infrastructure systems, such as database servers. Vulnerability in the IoT device communication protocol can spread to other devices that use the vulnerable protocol in the IoT infrastructure [6].

Thus, vulnerabilities in the protocols used in the IoT network can have devastating effects on the entire IoT infrastructure. The criticalities of these effects depend on the environments in which the compromised IoT devices operate.

Moreover, in some cases, the deployment conditions of IoT devices make it difficult or impossible to reconfigure or upgrade IoT devices. Often, IoT devices cannot be upgraded due to the discontinuation of device support from the manufacturer. This leads to the possibility of new vulnerabilities and threats to the IoT device in the future, as the current security mechanisms of device deployment may be out of date. Technical support and management of IoT smart devices are important cybersecurity issues in the long run. Another specific problem surrounding IoT cybersecurity is the fact that the internal operation of a smart device and the data streams generated by the device may be unknown to the user. The situation is complicated by the constant availability of IoT devices on a network and the ignorance of users (i.e., concerning potential cybersecurity risks). It may lead to the use of dangerous settings on IoT devices (default), direct network connections of internet devices to the internet, the use of obsolete or unreliable devices, and weak passwords.

One important IoT cybersecurity risk is that the functionality of smart devices can be changed by the device manufacturer without the consent or knowledge of the user (by updating the device firmware). It creates a new vulnerability that can allow the smart device to partially change the functionality or perform undesirable actions on the user's device, such as collecting sensitive user data without the user's knowledge.

However, the risks are not limited to data confidentiality. Attacks on IoT infrastructure can not only target compromised devices to steal sensitive data or cause financial losses but also disrupt or damage IoT devices physically. Compromised IoT devices can even lead to the injuries or deaths of people who depend on these devices or work with them.

Thus, non-compliance with basic security requirements (for both manufacturers and the users of smart devices) is the main cause of IoT cybersecurity problems. Common causes of security breaches in IoT infrastructure due to manufacturers are vulnerabilities in the IoT device software, lack of support for automatic updates, lack of firmware updates, and dangerous update mechanisms. This situation is often caused by manufacturers attempting to launch new smart devices as soon as possible. Vulnerabilities in software and web applications can lead to the theft of sensitive information or the spread of malicious firmware updates. Another common problem is unsafe authentication methods provided by the device manufacturers. The above weaknesses of the current IoT state of affairs, as well as the heterogeneity of the IoT environment, make IoT devices more vulnerable than computers and servers on conventional networks. Vulnerable components of IoT can be IoT devices, device software, and communication channels of the IoT infrastructure. The main threats in IoT infrastructure are distributed denial of service (DDoS), disclosure of confidential information, falsification, spoofing, and elevation of privilege. These threats are commonly used by cybercriminals as entry points, followed by other criminal activities: infecting devices with malicious software, stealing sensitive data, or blocking network connections.

Mentioned factors contribute to the high probability of compromising IoT devices, the spread of malicious software, and various multi-vector cyberattacks on IoT infrastructure (MVIA). At the same time, compromised IoT devices can be used as sources of attacks both inside and outside the IoT infrastructure.

The next subsection presents a brief analysis of the modern ideas and methods addressed to solve the problem of IoT malware detection by analyzing the advantages and disadvantages.

*1.2. Objectives and Contribution*

The main objectives of the work were to study the possibility of a multi-vector cyberattack detection in the IoT infrastructure based on a flow analysis and a deeper traffic analysis that takes into account IoT protocol features. This research aims to improve detection efficiency via various machine learning algorithm usages. The proposed approach deals with the set of network traffic features that may indicate the presence of cyberattacks in the IoT infrastructure and compromised IoT devices.

Thus, the novelty of this work involves the approach used for IoT multi-vector cyberattack detection, which involves a flow-based features analysis. It enables decreased detection time and is scalable. On the other hand, if the flow-based feature analysis was unable to conclude the attack presence, a deep analysis of network traffic with the use of MQTT-based, DNS-based, and HTTP-based features analysis was employed.

This paper is organized as follows. Section 2 presents the state-of-the-art. Section 3 describes the machine learning algorithms for cyberattack detection. Section 4 discusses the stages of the proposed IoT multi-vector cyberattack detection technique based on machine learning algorithms with the traffic features analysis. Section 5 proposes the experiments and the efficiency of the proposed approach. Finally, we present our conclusions and future research.

## 2. The State-of-the-Art

The scientific community is focusing on the increasing problems concerning cybersecurity today. Solutions devoted to cyberattack detection against Internet of Things infrastructure are widely presented [7,8]. Quite possibly, the most encouraging approaches for IoT cyberattack detection are based on machine learning algorithms (MLA) [9–13].

To solve the cyberattack detection problem, the authors of [14] proposed an approach that executes the IoT malware traffic analysis. It is based on the usage of multilevel artificial intelligence and involves neural networks and binary visualization. In addition, the approach proposes efficiency improvement via learning from the misclassification approach, which includes three main stages, is designed to collect the network traffic, perform the binary visualization to store the collected network traffic in ASCII, convert it to 2D images, and process/analyze the obtained binary image. An analysis of the binary images is executed using the TensorFlow tool, an end-to-end open-source platform designed to use machine learning for different problem solutions. It can find and classify patterns automatically. The main advantage of the tool is the ability to organize the system retraining as well as the possibility to make the image recognition. The approach proposes the use of the algorithm to perform the visualization of the collected traffic characteristics as an image (in the form of tiles using the Binvis tool). The TensorFlow machine tool can make predictions. The use of graphic tiles allows the determination of the tile combination on which the image is based. It is able to detect needed objects regardless of the location within the obtained image. The provided method can perform the IoT device protection on the gateway level, bypassing the IoT environment constraints.

The authors of [15] presented a survey on the experimental studies with a detailed analysis of a set of machine learning algorithms. The article included comparative data concerning the algorithm detection efficiency of anomalous behavior in IoT networks. Experimental results have shown that the best efficiency concerning used datasets is produced by the random forest algorithm. Nevertheless, all investigated machine learning algorithms demonstrated to be very close to random forest algorithm and detection efficiency results; sometimes the choice of an appropriate algorithm depends on the nature of the analyzed data.

Article [16] is devoted to machine learning classifiers involved in the botnet traffic analysis in the IoT environment. Nine IoT devices were employed for dataset construction, consisting of several botnet attack types. To evaluate the efficiency of the proposed approach, true positive, true negative, false positive, false negative, F1-score accuracy, precision, and recall were used. The experimental results of the research demonstrated that

the random forest algorithm produced the best results while the support vector machine produced the lowest results. The main disadvantage of the approach is the strong need for data analysis of all features in processed datasets.

The IoT cyberattack detection approach for the IoT network is presented in [17]. It is based on the use of intelligent technologies. The produced intelligent system operates with a set of network features. The approach aims to reduce the feature number via its ranking with the usage of the correlation coefficient, random forest algorithm, and the gain ratio. The base for the experimental research involves three feature sets, where using the proposed algorithm is to be combined to obtain an optimized feature set. The means of data processing the authors used were K-nearest neighbor, random forest, and XGBoost machine learning algorithms. All experiments were based on the usage of NSL-KDD, BoT-IoT, and DS2OS datasets. The investigation of the detection efficiency of the proposed system was executed. For this purpose, the metrics of accuracy, detection rate, F1-score, and precision were evaluated.

An approach for IoT attack detection based on the usage of cloud technologies and software-defined networks (SDNs) is presented in [18]. It employs a decentralized two-layer SDN and is able to perform attack mitigation in the wireless IoT infrastructure. To execute the network traffic control for each subnet domain, the predefined local domain controller of the specified domain was employed. The core of the approach is a special controller connected to a local controller and it is placed in the cloud environment. The approach also involves some special local controllers to perform the traffic collection from the investigated domains to perform the feature extraction, and, as a result, to find out the facts of the DDoS attack presence in the domain. The attack detection process is based on the analysis of 155 features, collected via the SPAN function of the Cisco switch. The obtained feature values were evaluated by detection modules placed within all defined local controllers to detect DDoS attacks. The approach used an extreme learning machine (ELM) as a decision-maker for attack detection. The feed-forward neural network with semi-supervised learning was used. The main advantage of ELM implementation is the training time reduction as it performs the random selection of the initial parameters. As a result, usage of ELM decreases the detection time. An attack mitigation module is also presented on each local controller. There is the possibility to organize the data exchange between each local controller, as well as with the universal controller. The proposed attack mitigation technique involves a set of attack mitigation scenarios able to perform in the wireless internet environment for different fixed devices.

The authors of [19] propose an intrusion detection system for IoT infrastructures. It is based on deep learning (DL-IDS). The approach for the IoT infrastructure intrusion detection involves the network traffic analysis; the data normalization procedure (to avoid the uncertainties in the obtained dataset); the data similarity evaluation on the usage of the Minkowski distance (to take into account the missing values, to eliminate possible redundancy, and to remove from the dataset the redundant and duplicate data); the replacement of the missing feature values in the obtained dataset (taking into account the evaluated values of the nearest neighbor on the basis of the K-nearest neighbor in the Euclidean distance to produce the average values for proceed data (to not take into account the classification results based on the data obtained from the more frequent entries); the traffic feature selection procedure on the basis of the spider monkey optimization algorithm usage (the set of features that are able to indicate the intrusion into the IoT infrastructure); and the exact intrusion detection procedure based on the stacked-deep polynomial network for the incoming data classification to mark it as normal or abnormal. The proposed approach is able to detect intrusions concerning the IoT environment (a remote-to-local attack, a DDoS attack, a probing attack, a user-to-root attack, etc.).

The study [20] provides research devoted to the usage of machine learning algorithms for anomaly detection in the Internet of Things infrastructures. To do this, the authors investigated the effectiveness and the main aspects of the usage of several single algorithms or their combinations for detection. The efficiency of the anomaly detection involved

performance metrics, such as false positives, false negatives, specificity, sensitivity, and overall accuracy. The experimental part of the study is based on the Nemenya and Friedman tests that made it possible to perform a statistical analysis of the classifiers' differences. Another aspect of the research was the evaluation of the classifiers' response time. For this purpose, specific IoT infrastructure (as part of the implemented IDS) was employed. As a result of the conducted experiments, the authors of the study concluded that the most acceptable classification accuracy and the time of response were provided by the classification trees, regression trees, and extreme gradient boosting.

An approach for cyberattack detection as an AD-IoT system is presented in [21]. The proposed system is designed for the smart city infrastructure and is based on the random forest machine learning algorithm. The system aims to detect the compromised IoT devices that are placed in the distributed fog nodes. The division of normal and malicious behaviors of IoT devices is executed on the basis of monitoring and analyzing the fog nodes' network traffic. Such analysis is performed to verify whether the fog level attacks are detected and to inform the cloud security services concerning the evaluated results. The presented approach demonstrates sufficient detection efficiency and applies to the smart city infrastructure.

An approach for DDoS attack detection is presented in [22]. It is based on the hybrid optimization algorithms of Metaheuristic lion and Firefly. It was designed to perform data collecting, data preprocessing for noise removing, and filling missing data. The feature extraction was performed by employing recursive feature elimination (RFE). An important item of the proposed technique is the possibility of detecting low-rate attacks using the hybrid ML-F optimization algorithm. For the attack classification, a random forest classifier was used.

The article [23] introduces an IDS, which is based on the technique that uses an ensemble-based voting classifier. This approach uses multiple classifiers as a base learner. The final prediction is formed via producing the classifier's vote for the traditional classifier predictions. As the mean of the efficiency evaluation of the presented approach, a set of IoT devices with the usage of different sensors (garage door, light motion, GPS sensor, fridge sensor, thermostat, modbus, and weather) were employed. Multi-class attacks, such as XSS, Ransomeware, scanning injection, DDoS, and backdoor, were involved in the technique efficiency verification. The efficiency of the presented method was compared with the set of new intrusion detection approaches provided by scientists. The comparison was constructed on the basis of the accuracy, precision, recall, and F-score metrics. Furthermore, a set of machine learning algorithms, such as decision tree, naive Bayes, random forest, and K-nearest neighbors were involved in the comparison procedure. The experimental results demonstrated that the proposed approach has a high detection efficiency.

The authors of [24] propose a detection method for DoS/DDoS attacks against the IoT using machine learning. The approach aims to detect and apply the mitigation scenarios in the situation of DoS/DDoS attacks. To do this, the approach employs a multiclass classifier ("Looking back"). In addition, the ability of the technique to detect "malicious" packets makes it possible to apply mitigation measures against attacks that employ specific packet types.

The approach in [25] provides a botnet detection system for IoT devices. It is based on the algorithm named local–global best bat, which is used for neural networks and is able to process the botnet's feature sets to distinguish malicious and benign network traffic. As an experimental part of the study, the botnets Mirai and Gafgyt were used to infect several commercial IoT devices. In addition, to classify 10 botnet classes, the proposed algorithm was used. It was designed to tune the neural network hyperparameters and optimize the weight. The authors made the efficiency comparison of the provided algorithm with other approaches. The experimental results demonstrated that the proposed botnet detection approach accuracy was up to 90%, while BA-NN was 85.5%, and PSO-NN was 85.2%.

The authors of [26] proposed a taxonomy of intrusions detection systems that utilizes the data objects as the dimensions to summarize and classify machine learning- and

deep learning-based IDS. The survey clarifies the concept of IDSs. Moreover, machine learning-based algorithms, metrics, and benchmark datasets frequently used in IDSs were introduced. IDSs applied to various data sources, i.e., logs, sessions, packets, and flow, were analyzed. The proposed taxonomic system was presented as a baseline and key IDS issues with using machine learning and deep learning algorithms. Moreover, future developments and challenges of IDS were discussed.

The authors of [27] introduced a probabilistic-driven ensemble (PDE)-based approach. This approach operates with several classification algorithms, wherein the effectiveness of these algorithms has been improved by applying a probabilistic criterion. Thus, the proposed approach allows maximizing the possibility of detecting intrusion events, regardless of the operational scenario, using several evaluation models. This makes it possible to distinguish ordinary events from related events to all classes of attacks. Experiments performed by using real-world data show that the proposed ensemble approach has better capability in detecting intrusion events (concerning known solutions).

The authors of [28] presented machine learning-based IDS. The feature reduction approach has two components: (1) Auto-encoder as a deep learning instance for dimensionality reduction; and (2) principal component analysis. The resulting set of low-dimensional features from both approaches was used to build different classifiers, i.e., Bayesian network, random forest, linear discriminant analysis, and quadratic discriminant analysis for designing IDS. The obtained experimental findings show better performance in terms of detection rate, false alarm rate, accuracy, and F-measure for binary and multi-class classification. This approach is able to reduce the feature dimensions of the CICIDS2017 dataset from 81 to 10, with high accuracy in both multi-class and binary classifications.

The objective of [29] was to apply various approaches for handling imbalanced datasets to design an effective IDS from the CIDDS-001 dataset. The effectiveness of sampling methods based on CIDDS-001 was studied and experimentally evaluated via random forest, deep neural networks, variational autoencoder, voting, and stacking machine learning classifiers. The developed system makes it possible to detect attacks with high accuracy when processing an unbalanced distribution of classes using a smaller number of samples. It makes it possible to apply the proposed system to data classification problems if it is necessary to merge data in real-time.

In [30], the authors were devoted to solving cybersecurity problems, such as the difficulty in distinguishing illegitimate activities from legitimate ones due to their high degrees of heterogeneity and similar characteristics. To solve this problem, a local feature engineering approach was proposed. This approach is based on the adoption of a data pre-processing strategy that allows reducing the number of network event patterns, increasing their characterization. The main distinguishing feature of the approach is that it operates locally in the feature space of each single network event, allowing to introduce new features and discretizing their values. The experimental results showed that the proposed approach improves the performance of known solutions.

The results of the machine learning algorithm efficiency analysis for detecting cyberattacks in the Internet of Things infrastructure are presented in Table 1.

The analysis of related works allows concluding that most studies had good detection accuracy; nevertheless, the main disadvantage of the investigated works is that they do not cover most features that may indicate the attack presence.

The analysis shows that the known approaches for detecting IoT cyberattacks demonstrate high-efficiency levels. Nevertheless, there are limitations—the inability to detect and respond to unknown attacks (zero-day attacks), the low efficiency of detection of multi-vector attacks; a high level of false positives, a significant response time that is unacceptable for real-time systems, and the need for significant amounts of computing resources. Another important aspect is the need to select a minimum and sufficient set of informative network traffic features that are able to indicate the presence of cyberattacks in the IoT infrastructure.

**Table 1.** Machine learning algorithm (MLA) efficiency for cyberattack detection in the Internet of Things infrastructure.

| Authors | Goal | MLA | Data Set | Result |
|---|---|---|---|---|
| Shire, R.; Shiaeles, S.; Bendiab, K.; Ghita B.; Kolokotronis, N. [14] | malware detection, zero-day malware classification | Convolutional Neural Network and binary visualization | Real network environments | Accuracy of 91.32%, Precision of 91.67%, Recall of 91.03% |
| Elmrabit, N.; Zhou, F.; Li, F.; Zhou H. [15] | anomaly detection, attack detection | Logistic Regression, Decision Tree, Adaptive boosting, KNN, Random Forest, Naive Bayes, Gated Recurrent Units, Simple Recurrent Neural Network, Convolutional Neural Network and Long short-Term Memory, Convolutional Neural Network, Long short-Term Memory, Deep Neural Network | UNSW-NB15, CICIDS-2017, ICS Cyberattack | Performance about 99.9% using Random Forest (CICIDS-2017) |
| Bagui, X. Wang; Bagui, S. [16] | intrusion detection | Logistic regression, SVM, random forest | UCI Machine Learning Repository | Accuracy of about 99% |
| Kumar, P.; Gupta, G.P.; Tripathi, R. [17] | cyber-attack detection against IoT networks | K-nearest neighbor, random forest, XGBoost | DS2OS, NSL-KDD, BoT-IoT | Accuracy up to 99%, detection 90–100% |
| Ravi N.; Shalinie S.M. [18] | DDoS attacks detection and attacks mitigation | ELM, semi-supervised extreme learning machines | UNB-ISCX | Accuracy of about 96.28% |
| Otoum, Y.; Liu, D.; Nayak A. [19] | DoS, user-to-root (U2R), remote-to-local (R2L) detection, probe, intrusions | Stacked-deep polynomial network | NSL-KDD | Accuracy up to 99.02%, Precision up to 99.4%, recall up to 98.3%, F1-score up to 98.8% |
| Verma, A.; Ranga, V. [20] | Survey on machine learning algorithms for DoS attacks detection | AdaBoost, extremely randomized trees, multilayer perceptron, classification and regression trees, random forest, gradient boosted machine, extreme gradient boosting | UNSW-NB15, NSL-KDD, CIDDS-001 | Regression trees, classification trees, and EG boosting show the best results—accuracy up to 96.7%, specificity up to 96.2%, sensitivity up to 97.3% |
| Alrashdi, I.; Alqazzaz, A.; Aloufi, E.; Alharthi, R.; Zohdy, M.; Ming, H. [21] | Detection of DDoS attacks | Bat Algorithm | N-BaIoT | Accuracy up to 90% |
| Krishna, E.S.; Thangavelu, A. [22] | Detection of the DDoS attacks | Random Forest | NSL-KDD, NBaIoT | Accuracy up to 99.98%, precision up to 99.87%, recall up to 100%, and F-score up to 99.73% |
| Mihoub, A.; Fredj, O.B.; Cheikhrouhou, O.; Derhab, A.; Krichen, M. [23] | Investigation of DoS/DDoS attacks detection for IoT based on ML algorithms | Looking-back-enabled random forest | IoT-Bot | Accuracy up to 99.81% |

**Table 1.** *Cont.*

| Authors | Goal | MLA | Data Set | Result |
|---|---|---|---|---|
| Khan, M.A.; Khan Khattk, M.A.; Latif, S.; Shah, A.A.; Ur Rehman, M.; Boulila, W.; Ahmad, J. [24] | intrusion detection | Combined decision tree, naive Bayes, random forest, and K-Nearest Neighbors using a voting-based technique | TON IoT | Accuracy up to 88%, Precision up to 90%, Recall up to 88%, F-score of 88% for DT-RF-NB based on binary classification with a combined IoT dataset |
| Alharbi, A.; Alosaimi, W.; Alyami, H.; Rauf, H.T. [25] | detection of DDoS attacks | Bat algorithm | N-BaIoT | Accuracy up to 90% |
| Saia, R.; Carta, S.; Recupero, D.R. [27] | intrusion events detection | Multilayer perceptron, decision tree, adaptive boosting, gradient boosting, random forests | NSL-KDD | Better performance compared to single classifiers in terms of specificity, without significant degradation in other aspects, since there is little degradation in terms of mean F-score, but a positive mean AUC (compared to competitor approaches), demonstrates the effectiveness of the approach |
| Abdulhammed, R.; Musafer, H.; Alessa, A.; Faezipour, M.; Abuzneid, A. [28] | developing the features dimensionality reduction approaches for machine learning-based IDS | Bayesian network, random forest, linear discriminant analysis, quadratic discriminant analysis | CICIDS2017 | Reducing the feature dimensions of a dataset from 81 to 10, with high accuracy of 99.6% in both multi-class and binary classification |
| Abdulhammed, R.; Faezipour, M.; Abuzneid, A.; AbuMallouh, A. [29] | applying various approaches for handling imbalanced datasets to design effective IDS | Random forest, deep neural networks, variational autoencoder, voting, stacking | CIDDS-001 | Attacks detection with up to 99.99% accuracy |
| Carta, S.; Podda, A.S.; Recupero, D.R.; Saia, R. [30] | solving such cybersecurity problems, as the difficulty of distinguishing illegitimate activities from legitimate ones | Random forests, decision tree, gradient boosting, adaptive boosting, multilayer perceptron | NSL-KDD, CICIDS2017, UNSW-NB15 | Improving the performance of the state-of-the-art canonical solutions |

To summarize, there is a strong need to evolve new methods for cyberattack detection in the IoT infrastructure. To do this, we are to eliminate technique drawbacks and increase the detection efficiency of detecting known and unknown cyberattacks in the IoT infrastructure.

## 3. Machine Learning Algorithms for Cyberattack Detection

The current study has involved five MLAs for IoT multi-vector cyberattack detections, as they were mostly used in (recent) research for efficient object classification [15–17,20,22,30]; we relied on our own experience in MLA use for cyberattack detection [11]:

1.  Decision tree (DT) [31,32];
2.  Random forest (RF) [33–38];
3.  K-Nearest Neighbor (KNN) [39];
4.  Extreme Gradient Boosting (XGBoost) [40];
5.  Support Vector Machine (SVM) [41–43].

## 4. IoT Multi-Vector Cyberattack Detection Based on Machine Learning Algorithms

### 4.1. Detection Steps

The approach for IoT cyberattack detection includes the following steps (Figure 1):

1.  Traffic obtaining;
2.  Grouping packets by type, source device, and time. Packets from each device are grouped by type and by N records, according to the last connection time;
3.  Feature extraction;
4.  Feature classification based on the machine learning algorithm;
5.  Result producing.

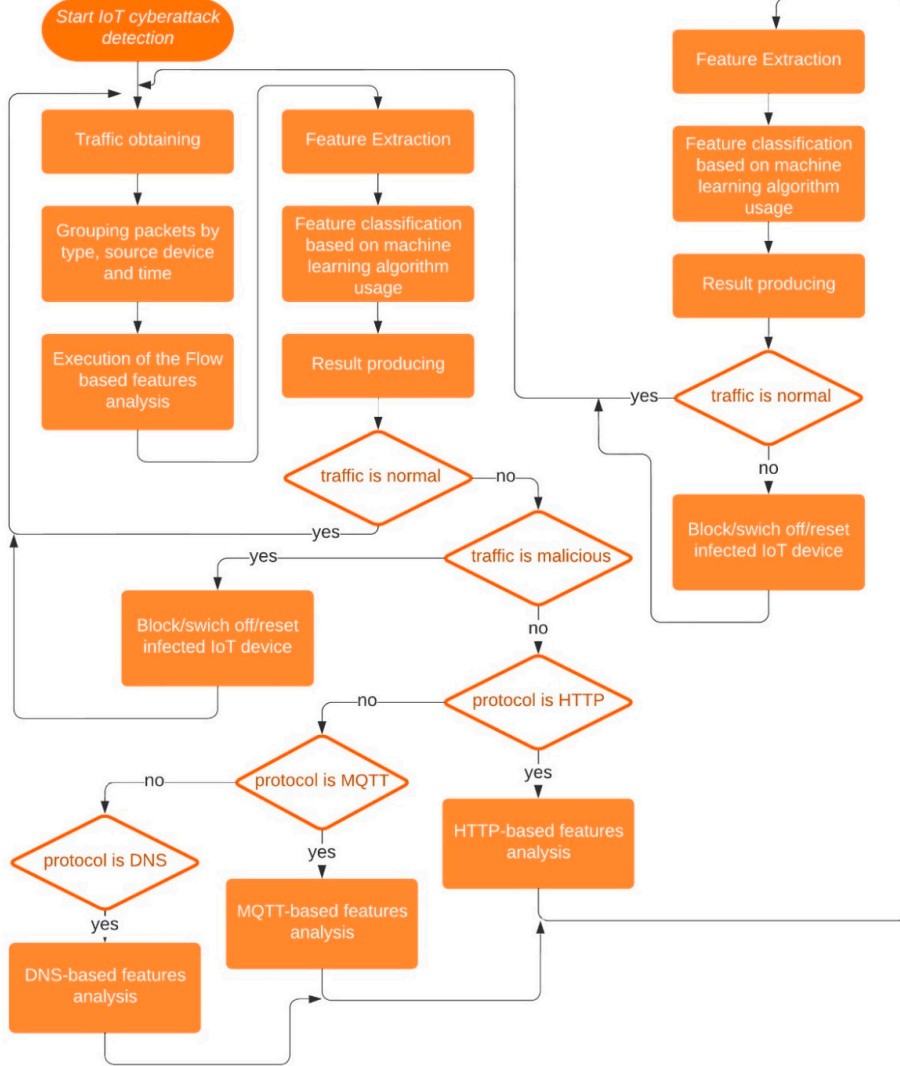

**Figure 1.** IoT cyberattack detection scheme.

### 4.2. Features Description

An important task is to speed up the detection of attack traffic. Early detection of attack traffic provides an opportunity to increase the security of the Internet of Things infrastructure, as it prevents the further spread of malicious software compromising not yet infected devices in the IoT infrastructure. Therefore, to speed up the detection of cyberattacks in the infrastructure, four types of features are involved:

- Flow-based features;
- MQTT-based features;
- DNS-based features;
- HTTP-based features.

Using only flow-based features (Table 2) makes it possible to speed up the detection of attacks on the network by faster extraction of features from streams and their analyses. In the case of suspicious traffic behavior that cannot be unambiguously classified as an attack, an in-depth traffic analysis is applied with the MQTT-based (Table 3), DNS-based (Table 4), and HTTP-based (Table 5) feature extractions.

**Table 2.** Flow-based features.

| # | Features Designation | Value Description |
|---|---|---|
| 1 | $f_1$ | Protocol type |
| 2 | $f_2$ | Source IP address |
| 3 | $f_3$ | Destination IP address |
| 4 | $f_4$ | Source port |
| 5 | $f_5$ | Destination port |
| 6 | $f_6$ | Last connection time |
| 7 | $f_7$ | Transaction bytes from f2 to f3 |
| 8 | $f_8$ | Transaction bytes from f3 to f2 |
| 9 | $f_9$ | Mean packet size transmitted by f2 |
| 10 | $f_{10}$ | Mean packet size transmitted by f3 |
| 11 | $f_{11}$ | Source bits per second |
| 12 | $f_{12}$ | TTL value, f2 to f3 |
| 13 | $f_{13}$ | TTL value, f3 to f2 |
| 14 | $f_{14}$ | Interpacket interval |
| 15 | $f_{15}$ | Bandwidth |
| 16 | $f_{16}$ | Packet jitter |

**Table 3.** MQTT-based features.

| # | Features Designation | Value Description |
|---|---|---|
| 1 | $f_{18}$ | The amount of connections to $f_3$ in N gathered records according to $f_6$ |
| 2 | $f_{19}$ | The amount of connections of $f_2$ in N gathered records according to $f_6$ |
| 3 | $f_{20}$ | The amount of connections of $f_2$ and $f_5$ in N gathered records according to $f_6$ |
| 4 | $f_{21}$ | The amount of connections to $f_3$ and $f_4$ in N gathered records according to $f_6$ |
| 5 | $f_{22}$ | The amount of connections of $f_2$ and $f_3$ in N gathered records according to $f_6$ |

**Table 4.** HTTP-based features.

| # | Features Designation | Value Description |
|---|---|---|
| 1 | $f_{48}$ | HTTP request method (GET, POST, HEAD) |
| 2 | $f_{49}$ | HOST header value |
| 3 | $f_{50}$ | Length of the HOST header value |
| 4 | $f_{51}$ | URL in the request |
| 5 | $f_{52}$ | Length of URL |
| 6 | $f_{53}$ | HTTP pipelining depth |
| 7 | $f_{54}$ | Uncompressed size of the transferred data from the client |

**Table 4.** *Cont.*

| # | Features Designation | Value Description |
|---|---|---|
| 8 | $f_{55}$ | Uncompressed size of the transferred data from the server |
| 9 | $f_{56}$ | Percentage of $f_{48}$ with the same $f_{49}$ in N records according to $f_6$ |
| 10 | $f_{57}$ | Percentage of the $f_{49}$ the with same the $f_{51}$ in N records according to $f_6$ |
| 11 | $f_{58}$ | Percentage of $f_{48}$ with the same $f_{51}$ in N records according to $f_6$ |

**Table 5.** DNS-based features.

| # | Features Designation | Value Description |
|---|---|---|
| 1 | $f_{23}$ | Requested domain name |
| 2 | $f_{24}$ | Value specifying the request type |
| 3 | $f_{25}$ | Length of $f_{23}$ |
| 4 | $f_{26}$ | Amount of unique characters in $f_{23}$ |
| 5 | $f_{27}$ | Entropy of $f_{23}$ |
| 6 | $f_{28}$ | TTL-period, mode (the value that appears most often in a set of data), in N records according to $f_6$ |
| 7 | $f_{29}$ | TTL-period, median (the numerical value separating the higher half of a data sample from the lower half), in N records according to $f_6$ |
| 8 | $f_{30}$ | TTL-period, average value, in N records according to $f_6$ |
| 9 | $f_{31}$ | Amount of A-records corresponding to $f_{23}$ in the incoming DNS messages (the feature is used if value $f_{31} > 1$), in N records according to $f_6$ |
| 10 | $f_{32}$ | Amount of IP addresses concerned with $f_{23}$ (feature is used if value $f_{31} = 1$), in N records according to $f_6$ |
| 11 | $f_{33}$ | Average distance between the IP addresses concerned with $f_{23}$ (feature is used if value $f_{31} = 1$), in N records according to $f_6$ |
| 12 | $f_{34}$ | Average distance between the IP addresses in the set of A-records for $f_{23}$ in the incoming DNS message (feature is used if value $f_{31} > 1$), in N records according to $f_6$ |
| 13 | $f_{35}$ | Amount of unique IP addresses in sets of A-records corresponding to $f_{23}$ in the DNS messages (feature is used if value $f_{31} > 1$), in N records according to $f_6$ |
| 14 | $f_{36}$ | Average distance between unique IP addresses in sets of A-records corresponding to $f_{23}$ in the DNS messages (feature is used if value $f_{31} > 1$), in N records according to $f_6$ |
| 15 | $f_{37}$ | Domain name amounts that share IP addresses corresponding to $f_{23}$, in N records according to $f_6$ |
| 16 | $f_{38}$ | Sign of the usage of uncommon types of DNS records, or DNS records that are not commonly used by a typical client (e.g., TXT are most often used for tunneling (excluding mail servers), KEY, or NULL) |
| 17 | $f_{39}$ | The entropy of the DNS records, which are contained in the DNS messages (CNAME, TXT, NS, MX, KEY, NULL, etc.) |
| 18 | $f_{40}$ | Maximum size of the DNS messages about $f_{23}$, in N records according to $f_6$ |
| 19 | $f_{41}$ | Sign of success of DNS query ($f_{41} = 0$ if DNS query failed, and $f_{41} = 1$ if DNS query was successful) |
| 20 | $f_{42}$ | Answer length |
| 21 | $f_{43}$ | Mean class value in N records according to $f_6$ |
| 22 | $f_{44}$ | Mean type value in N records according to $f_6$ |
| 23 | $f_{45}$ | Amount of $f_2$ and $f_{23}$ in N records according to $f_6$ |
| 24 | $f_{46}$ | Amount of $f_{23}$ to the same $f_2$ in N records according to $f_6$ |
| 25 | $f_{47}$ | Percentage of the domain in N records according to $f_6$ |

This section presents the involvement of four feature types for multi-vector cyberattack detection in the IoT infrastructure. The features based on flow analysis enable the possibility of speeding up attack detections through faster analyses and make the detection algorithm scalable, allowing us to analyze high-bandwidth IoT traffic. On the other hand, the features based on deep packet analyses enable us to improve the accuracy of detection in cases where the use of a sign based on flow analysis does not provide an unambiguous answer about the presence of a cyberattack (and also allows detecting the multi-vector attacks).

## 5. Experiments

### 5.1. Evaluation Setting

To conduct the experiments, a Wi-Fi network of IoT devices was created. A Raspberry Pi 3 was configured as a middlebox, which acted as a Wi-Fi access point. To simulate DoS attacks as a source of malicious traffic, a computer system with a virtual Kali Linux was used. As a victim of DoS attacks, Raspberry Pi 2 with an installed Apache web server was used. All devices were connected to create a Wi-Fi network access point.

Three IoT devices (router, thermostat, camcorder) were also connected to the Wi-Fi network. To obtain normal traffic, a simulation of user interactions with the devices of the created IoT network was performed. To do this, actions such as transmitting video from the camera and installing software updates on connected IoT devices were performed. To obtain malicious traffic, a simulation of performing the most common classes of DoS attacks was executed.

An HTTP GET flood attack was simulated with the Goldeneye tool [44]; TCP SYN and UDP flood were simulated with Kali Linux hping3 utility [45]. The iodine utility was used to perform DNS tunneling attacks [46].

Malicious/benign traffic was collected at the Wi-Fi access point. The IoT traffic collection was executed via the Zeek tool [47]. It gives capacities to the network intrusion detection systems (IDS) and empowers security operation centers (SOC). The Zeek tool was used as a network traffic analyzer with an in-built classification engine.

In the collected DoS traffic samples, the source IP addresses and MAC addresses were substituted for the IP addresses and MAC addresses of the devices of the created IoT network. The time of sending malicious packets was modified so that the total collected IoT traffic replicated the activity of the attacking and normal activity devices.

Thus, the execution of DoS attacks of different types by each IoT device was simulated.

### 5.2. Dataset Description

To hold the experiments, the traffic generated by Mirai, Gafgyt, Dark Nexus botnets, UCI Machine Learning Repository, DS2OS, Bot-IoT, N-BaIoT, CIDDS, UNSW-NB15, and NSL-KDD traffic datasets [48–54] were used.

The DS2OS dataset contains traces gathered from the application layer of the IoT environment from devices such as movement sensors, light controllers, thermometers, batteries, thermostats, smart doors, etc. This dataset can be used to assess anomaly-based attack detection algorithms.

The UNSW-NB15 dataset contains data on nine types of attacks, such as Fuzzers, Analysis, Backdoors, DoS, Exploits, Generic, Reconnaissance, Shellcode, and Worms. A total of 49 features were extracted to describe these types of attacks.

The N_BaIoT dataset offers real-world IoT traffic data collected from nine IoT devices infected by Mirai and BASHLITE. Malicious data are divided into 10 attacks as well as harmless data (with 115 different features).

The Kitsune Network Attack Dataset contains nine network capture datasets in total that relate to different types of attack traffic against the IoT Infrastructure.

The BoT-IoT dataset was created by deploying a realistic IoT infrastructure network environment and it includes legitimate IoT network traffic as well as various types of attacks. The BoT-IoT includes DDoS and DoS for different protocols, OS scan, service scan, data exfiltration, and keylogging attacks.

The CIDDS and NSL-KDD datasets are built on network intrusion data describing "bad" connections, which are called intrusions (or attacks) and "good" connections (legitimate connections). These databases describe a wide range of intrusions and take into account user behavior scenarios.

Furthermore, experiments dealt with the set of traffic features presented in the above-mentioned datasets for three IoT devices: router, thermostat, and camcorder that were infected by Mirai, Gafgyt, and Dark Nexus botnets. The set of traffic features corresponds to four types of attacks (TCP, UDP, HTTP GET, and DNS tunneling).

As each dataset contains different samples and features, the preprocessing and feature selection processes were executed via each file type analysis and their parsing into the needed presentation for the next preprocessing. Such files as .csv, .pcap, Argus files, Zeek files, and .txt were processed.

Mirai is well-known malware that is able to infect an IoT device and turn such a smart device into a remotely-controlled network of bots—a botnet. The main negative impact of Mirai is the ability to launch massive DDoS attacks, as well as the ability to scan the internet for IoT smart devices based on the ARC processor. Such vulnerability as the usage of a stripped-down Linux version makes it possible to perform the logging into the device and execute malicious actions. In addition, the Mirai botnet uses a great amount of hijacked IoT devices to increase its spread and it is very dangerous for its mutating [55].

Gafgyt is a botnet that uses the vulnerabilities of IoT devices. It employs infected devices for large-scale (DDoS) attack execution. Moreover, Gafgyt uses known vulnerabilities (e.g., CVE-2017-17215, CVE-2018-10561) to implement the downloading of the next-stage payloads to compromised devices. New versions of the Gafgyt botnet include Mirai-based components to perform DDoS attacks; HTTP flooding to send a great number of HTTP requests to server targets to overwhelm them; UDP flooding to send special UDP packets to server victims to exhaust them; TCP flood attacks; STD attacks to send a random string to a specified IP address [56].

Dark Nexus is an IoT botnet that launches DDoS attacks. It was designed to launch credential stuffing attacks against different kinds of IoT devices (video recorders; DLink, Dasan Zhone, ASUS routers, thermal cameras, etc.) [57].

### 5.3. Training and Testing

The proposed approach involves five ML algorithms (decision tree, random forest, K-nearest neighbor, extreme gradient boosting, and support vector machine) to compare their detection possibilities. All algorithms were trained and tested using the dataset with training and testing percentages of 75% and 25%.

The BotGRABBER framework uses the scikit-learn library–an open-source platform for MLA in Python [58]. The configuration of each used MLA relies on the appropriate set of algorithm parameters. The optimal used values of algorithm parameters are presented in Tables 6–10 [59–63].

**Table 6.** Decision tree algorithm parameters [59].

| Parameter | Value | Description |
| --- | --- | --- |
| criterion | gini | The function to measure the quality of a split. |
| splitter | best | The strategy used to choose the split at each node. |
| max_depth | None | The maximum depth of the tree. |
| min_samples_split | 3 | The minimum number of samples required to split an internal node. |
| min_samples_leaf | 1 | The minimum number of samples required to be at a leaf node. |
| min_weight_fraction_leaf | 0.0 | The minimum weighted fraction of the sum total of weights (of all the input samples) required to be at a leaf node. |
| max_features | auto | The number of features to consider when looking for the best split. |
| random_state | RandomState instance | Controls the randomness of the estimator. |
| class_weight | balanced | Weights associated with classes. |
| ccp_alpha | 0.0 | Complexity parameter used for minimal cost complexity pruning. |

**Table 7.** Random forest algorithm parameters [60].

| Parameter | Value | Description |
| --- | --- | --- |
| n_estimators | 100 | The number of trees in the forest. |
| criterion | gini | The function to measure the quality of a split. |
| max_depth | None | The maximum depth of the tree. |
| min_samples_split | 2 | The minimum number of samples required to split an internal node. |

**Table 7.** *Cont.*

| Parameter | Value | Description |
|---|---|---|
| min_samples_lea | 1 | The minimum number of samples required to be at a leaf node. |
| min_weight_fraction_leaf | 0.0 | The minimum weighted fraction of the sum total of weights. |
| max_features | log$^2$ | The number of features to consider when looking for the best split. |
| class_weight | balanced | Weights associated with classes. |
| ccp_alpha | 0.0 | Complexity parameter used for minimal cost complexity pruning. |

**Table 8.** K-Nearest Neighbor algorithm parameters [61].

| Parameter | Value | Description |
|---|---|---|
| n_neighbors | 5 | Number of neighbors. |
| weights | distance | Weight function used in prediction. |
| algorithm | kd_tree | The algorithm used to compute the nearest neighbors. |
| leaf_size | 30 | Leaf size passed to KDTree. |
| p | 2 | Power parameter for the Minkowski metric. |
| metric | str | The distance metric to use for the tree. |
| metric_params | dict | The number of parallel jobs to run for the neighbors' search. |

**Table 9.** Extreme gradient boosting algorithm parameters [62].

| Parameter | Value | Description |
|---|---|---|
| loss | exponential | The loss function to be optimized. |
| learning_rate | 0.1 | Learning rate shrinks the contribution of each tree. |
| n_estimators | 100 | The number of boosting stages to perform. |
| subsample | 1.0 | The fraction of samples to be used for fitting the individual base learners. |
| criterion | squared_error | The function to measure the quality of a split. |
| min_samples_split | 2 | The minimum number of samples required to split an internal node. |
| min_weight_fraction_leaf | 0.0 | The minimum weighted fraction of the sum total of weights (of all the input samples) required to be at a leaf node. |
| max_depth | 3 | The maximum depth of the individual regression estimator. |
| random_state | RandomState instance | Controls the random seed given to each tree estimator at each boosting iteration. |
| max_features | None | The number of features to consider when looking for the best split. |
| max_leaf_nodes | None | Grow trees with max_leaf_nodes in the best-first fashion. |
| validation_fraction | 0.1 | The proportion of training data to set aside as the validation set for early stopping. |
| n_iter_no_change | None | The decision as to whether early stopping will be used to terminate training when the validation score does not improve. |
| tol | $1 \times 10^3$ | Tolerance for the early stopping. |
| ccp_alpha | 0.0 | Complexity parameter used for minimal cost complexity pruning. |

**Table 10.** Support vector machine parameters [63].

| Parameter | Value | Description |
|---|---|---|
| C | 1.0 | Regularization parameter. |
| kernel | rbf | Specifies the kernel type to be used in the algorithm. |
| gamma | auto | Kernel coefficient. |
| tol | $1 \times 10^3$ | Tolerance for stopping criterion. |
| cache_size | 100 | Specify the size of the kernel cache (in MB). |
| max_iter | −1 | Hard limit on iterations (no limit). |
| random_state | RandomState instance | Controls the pseudo-random number generation to shuffle the data for probability estimates. |

### 5.4. Implementation Platform

To perform the feature extraction, the feature classification based on the machine learning algorithm, as well as the result of production, the BotGRABBER framework was employed. It is a multi-vector protection system that can perform network and host activity analyses. The BotGRABBER framework presents the tool, not only for botnet detection but also to produce the needed security scenario of the network reconfiguration according to the type of cyberattack performed by the detected botnet [11,13,43]. The mentioned tool includes several units aimed at traffic collection, packet processing, feature extraction, feature classification based on machine learning algorithms, and producing results. The feature classification unit of the framework is based on the scikit-learn library usage. It is a free software ML library for the Python programming language [58].

### 5.5. Results

Experimental results are presented in Tables 11–19.

As examples, comparisons of the different MLA efficiencies for Router/Mirai botnet detection (TCP attack, UDP attack, HTTP GET attack, and DNS tunneling) are presented in Figures 2–4.

**Table 11.** Classification results (router—Mirai).

| Device/ Botnet | Attack | Algorithm | Accuracy | TP | FP | FN | TN | Precision | Recall | F1 Score | AUC |
|---|---|---|---|---|---|---|---|---|---|---|---|
| Router/ Mirai | TCP | RF | 0.999479 | 3620 | 2 | 4 | 2024 | 0.9994748 | 0.999896 | 0.999572 | 0.999615 |
| | | DT | 0.998584 | 3612 | 3 | 5 | 2030 | 0.99917 | 0.998618 | 0.998894 | 0.998994 |
| | | kNN | 0.999469 | 3603 | 1 | 2 | 2044 | 0.999723 | 0.999445 | 0.999584 | 0.999692 |
| | | XGBoost | 0.998938 | 3562 | 5 | 1 | 2082 | 0.998598 | 0.999719 | 0.999158 | 0.999573 |
| | | SVM | 0.996991 | 3544 | 6 | 11 | 2089 | 0.99831 | 0.996906 | 0.997607 | 0.997881 |
| | UDP | RF | 0.999767 | 7531 | 5 | 2 | 2012 | 0.999937 | 0.999835 | 0.999935 | 0.999841 |
| | | DT | 0.999267 | 7515 | 4 | 3 | 2028 | 0.999468 | 0.999601 | 0.999534 | 0.99975 |
| | | kNN | 0.999476 | 7470 | 2 | 3 | 2075 | 0.999732 | 0.999599 | 0.999665 | 0.999821 |
| | | XGBoost | 0.999686 | 7465 | 1 | 2 | 2082 | 0.999866 | 0.999732 | 0.999799 | 0.999827 |
| | | SVM | 0.998534 | 7455 | 10 | 17 | 2068 | 0.998678 | 0.998678 | 0.998678 | 0.999174 |
| | HTTP GET | RF | 0.999694 | 6434 | 3 | 3 | 2060 | 0.999834 | 0.999734 | 0.999734 | 0.999839 |
| | | DT | 0.999412 | 6419 | 1 | 4 | 2076 | 0.999844 | 0.999377 | 0.999611 | 0.999793 |
| | | kNN | 0.999412 | 6387 | 1 | 4 | 2108 | 0.999843 | 0.999374 | 0.999609 | 0.999458 |
| | | XGBoost | 0.999529 | 6340 | 2 | 2 | 2156 | 0.999685 | 0.999685 | 0.999685 | 0.999671 |
| | | SVM | 0.997412 | 6381 | 5 | 14 | 2100 | 0.998636 | 0.99637 | 0.997502 | 0.999051 |
| | DNS tunneling | RF | 0.999624 | 5978 | 3 | 4 | 2005 | 0.999798 | 0.999731 | 0.999615 | 0.999944 |
| | | DT | 0.999249 | 5935 | 2 | 4 | 2049 | 0.999663 | 0.999326 | 0.999495 | 0.999928 |
| | | kNN | 0.999374 | 5920 | 3 | 2 | 2065 | 0.999493 | 0.999662 | 0.999578 | 0.999632 |
| | | XGBoost | 0.998999 | 5903 | 5 | 3 | 2079 | 0.999154 | 0.999492 | 0.999323 | 0.999186 |
| | | SVM | 0.997247 | 5899 | 5 | 14 | 2072 | 0.998649 | 0.99542 | 0.997032 | 0.997547 |

**Table 12.** Classification results (router—Gafgyt).

| Device/ Botnet | Attack | Algorithm | Accuracy | TP | FP | FN | TN | Precision | Recall | F1 Score | AUC |
|---|---|---|---|---|---|---|---|---|---|---|---|
| Router/ Gafgyt | TCP | RF | 0.999714 | 11984 | 2 | 2 | 2002 | 0.999833 | 0.999833 | 0.999833 | 0.999835 |
| | | DT | 0.999571 | 11963 | 2 | 4 | 2021 | 0.999833 | 0.999666 | 0.999749 | 0.999757 |
| | | kNN | 0.999357 | 11917 | 4 | 5 | 2064 | 0.999664 | 0.999581 | 0.999623 | 0.999792 |
| | | XGBoost | 0.999643 | 11881 | 3 | 2 | 2104 | 0.999748 | 0.999832 | 0.99979 | 0.999734 |
| | | SVM | 0.998713 | 11888 | 7 | 11 | 2084 | 0.999412 | 0.999076 | 0.999244 | 0.999523 |
| | UDP | RF | 0.999738 | 4498 | 2 | 1 | 1999 | 0.999656 | 0.999878 | 0.999667 | 0.999882 |
| | | DT | 0.999077 | 4453 | 4 | 2 | 2041 | 0.999103 | 0.999551 | 0.999327 | 0.99947 |
| | | kNN | 0.999385 | 4430 | 3 | 1 | 2066 | 0.999323 | 0.999774 | 0.999549 | 0.999648 |
| | | XGBOOST | 0.999077 | 4391 | 5 | 1 | 2103 | 0.998863 | 0.999772 | 0.999317 | 0.999712 |
| | | SVM | 0.998308 | 4433 | 6 | 9 | 4433 | 0.999056 | 0.998867 | 0.998961 | 0.998861 |
| | HTTP GET | RF | 0.999784 | 21082 | 2 | 3 | 2013 | 0.999905 | 0.999858 | 0.999881 | 0.999913 |
| | | DT | 0.99987 | 21034 | 1 | 2 | 2063 | 0.999952 | 0.999905 | 0.999929 | 0.999912 |
| | | kNN | 0.999697 | 20997 | 2 | 5 | 2096 | 0.999905 | 0.999762 | 0.999833 | 0.999971 |
| | | XGBoost | 0.999827 | 20990 | 1 | 3 | 2106 | 0.999952 | 0.999857 | 0.999905 | 0.999845 |
| | | SVM | 0.998961 | 20986 | 6 | 17 | 18684 | 0.998409 | 0.996144 | 0.997275 | 0.999221 |
| | DNS tunneling | RF | 0.999846 | 3191 | 2 | 4 | 2003 | 0.999674 | 0.999748 | 0.999561 | 0.999783 |
| | | DT | 0.998269 | 3153 | 5 | 4 | 2038 | 0.998417 | 0.998733 | 0.998575 | 0.999548 |
| | | kNN | 0.998654 | 3115 | 2 | 5 | 2078 | 0.999358 | 0.998397 | 0.998878 | 0.999539 |
| | | XGBoost | 0.999615 | 3074 | 1 | 1 | 2124 | 0.999675 | 0.999675 | 0.999675 | 0.999882 |
| | | SVM | 0.996154 | 3121 | 10 | 11 | 1485 | 0.998919 | 0.995688 | 0.997301 | 0.997861 |

**Table 13.** Classification results (router—Dark Nexus).

| Device/ Botnet | Attack | Algorithm | Accuracy | TP | FP | FN | TN | Precision | Recall | F1 Score | AUC |
|---|---|---|---|---|---|---|---|---|---|---|---|
| Router/ Dark nexus | TCP | RF | 0.999333 | 5490 | 4 | 1 | 2005 | 0.999272 | 0.999818 | 0.999545 | 0.999691 |
| | | DT | 0.9992 | 5472 | 5 | 1 | 2022 | 0.999087 | 0.999817 | 0.999452 | 0.999982 |
| | | kNN | 0.998933 | 5455 | 3 | 5 | 2037 | 0.99945 | 0.999084 | 0.999267 | 0.999836 |
| | | XGBOOST | 0.9992 | 5417 | 2 | 4 | 2077 | 0.999631 | 0.999262 | 0.999446 | 0.999285 |
| | | SVM | 0.9976 | 5394 | 9 | 9 | 2088 | 0.998334 | 0.998334 | 0.998334 | 0.999444 |
| | UDP | RF | 0.999344 | 10196 | 5 | 3 | 1996 | 0.99951 | 0.999706 | 0.999608 | 0.999488 |
| | | DT | 0.999672 | 10171 | 1 | 3 | 2025 | 0.999902 | 0.999705 | 0.999803 | 0.999932 |
| | | kNN | 0.999426 | 10146 | 3 | 4 | 2047 | 0.999704 | 0.999606 | 0.999655 | 0.999835 |
| | | XGBOOST | 0.999426 | 10120 | 2 | 5 | 2073 | 0.999802 | 0.999506 | 0.999654 | 0.999844 |
| | | SVM | 0.998279 | 10137 | 7 | 9 | 10137 | 0.998301 | 0.997736 | 0.998019 | 0.998421 |
| | HTTP GET | RF | 0.999771 | 19767 | 2 | 3 | 2018 | 0.999899 | 0.999848 | 0.999874 | 0.999853 |
| | | DT | 0.999725 | 19746 | 3 | 3 | 2038 | 0.999848 | 0.999848 | 0.999848 | 0.999995 |
| | | kNN | 0.999679 | 19716 | 2 | 5 | 2067 | 0.999899 | 0.999746 | 0.999823 | 0.999931 |
| | | XGBOOST | 0.999771 | 19666 | 1 | 4 | 2119 | 0.999949 | 0.999797 | 0.999873 | 0.999794 |
| | | SVM | 0.99899 | 19665 | 4 | 18 | 19665 | 0.99909 | 0.995918 | 0.997502 | 0.999452 |
| | DNS tunneling | RF | 0.999298 | 9351 | 3 | 5 | 2041 | 0.999679 | 0.999466 | 0.999572 | 0.999457 |
| | | DT | 0.999474 | 9301 | 4 | 2 | 2093 | 0.99957 | 0.999785 | 0.999678 | 0.999974 |
| | | kNN | 0.999737 | 9285 | 2 | 1 | 2112 | 0.999785 | 0.999892 | 0.999838 | 0.999859 |
| | | XGBOOST | 0.999386 | 9243 | 5 | 2 | 2150 | 0.999459 | 0.999784 | 0.999621 | 0.999482 |
| | | SVM | 0.997895 | 9302 | 7 | 10 | 9302 | 0.998109 | 0.99542 | 0.996763 | 0.998561 |

**Table 14.** Classification results (thermostat—Mirai).

| Device/ Botnet | Attack | Algorithm | Accuracy | TP | FP | FN | TN | Precision | Recall | F1 Score | AUC |
|---|---|---|---|---|---|---|---|---|---|---|---|
| Thermostat/ Mirai | TCP | RF | 0.999938 | 3623 | 1 | 5 | 2021 | 0.999724 | 0.999622 | 0.999773 | 0.999913 |
| | | DT | 0.998938 | 3618 | 1 | 5 | 2026 | 0.999724 | 0.99862 | 0.999171 | 0.999446 |
| | | kNN | 0.998938 | 3569 | 5 | 1 | 2075 | 0.998601 | 0.99972 | 0.99916 | 0.999017 |
| | | XGBOOST | 0.999646 | 3528 | 1 | 1 | 2120 | 0.999717 | 0.999717 | 0.999717 | 0.999923 |
| | | SVM | 0.996106 | 3535 | 5 | 17 | 2093 | 0.998588 | 0.995214 | 0.996898 | 0.999678 |
| | UDP | RF | 0.999986 | 7495 | 2 | 1 | 2052 | 0.999933 | 0.999897 | 0.9999 | 0.999865 |
| | | DT | 0.999791 | 7451 | 1 | 1 | 2097 | 0.999866 | 0.999866 | 0.999866 | 0.999834 |
| | | kNN | 0.999372 | 7446 | 1 | 5 | 2098 | 0.999866 | 0.999329 | 0.999597 | 0.999701 |
| | | XGBOOST | 0.999476 | 7407 | 1 | 4 | 2138 | 0.999865 | 0.99946 | 0.999663 | 0.999991 |
| | | SVM | 0.997906 | 7446 | 3 | 10 | 7446 | 0.999056 | 0.997172 | 0.998113 | 0.999816 |
| | HTTP GET | RF | 0.999859 | 6438 | 4 | 4 | 2054 | 0.999879 | 0.999779 | 0.999779 | 0.999861 |
| | | DT | 0.999529 | 6391 | 2 | 2 | 2105 | 0.999687 | 0.999687 | 0.999687 | 0.999722 |
| | | kNN | 0.999412 | 6369 | 4 | 1 | 2126 | 0.999372 | 0.999843 | 0.999608 | 0.999893 |
| | | XGBOOST | 0.999294 | 6343 | 5 | 1 | 2151 | 0.999212 | 0.999842 | 0.999527 | 0.999791 |
| | | SVM | 0.997176 | 6402 | 9 | 10 | 6402 | 0.998409 | 0.996144 | 0.997275 | 0.999465 |
| | DNS tunneling | RF | 0.999649 | 5976 | 1 | 5 | 2008 | 0.999833 | 0.999864 | 0.999498 | 0.999692 |
| | | DT | 0.998874 | 5967 | 4 | 5 | 2014 | 0.99933 | 0.999163 | 0.999246 | 0.999617 |
| | | kNN | 0.999249 | 5925 | 3 | 3 | 2059 | 0.999494 | 0.999494 | 0.999494 | 0.999828 |
| | | XGBOOST | 0.999374 | 5876 | 4 | 1 | 2109 | 0.99932 | 0.99983 | 0.999575 | 0.999422 |
| | | SVM | 0.996996 | 5890 | 10 | 15 | 5890 | 0.998379 | 0.995152 | 0.996763 | 0.998059 |

**Table 15.** Classification results (thermostat—Gafgyt).

| Device/ Botnet | Attack | Algorithm | Accuracy | TP | FP | FN | TN | Precision | Recall | F1 Score | AUC |
|---|---|---|---|---|---|---|---|---|---|---|---|
| Thermostat/ Gafgyt | TCP | RF | 0.999943 | 11973 | 3 | 2 | 2012 | 0.999849 | 0.999833 | 0.999891 | 0.999954 |
| | | DT | 0.999714 | 11942 | 2 | 2 | 2044 | 0.999833 | 0.999833 | 0.999833 | 0.999876 |
| | | kNN | 0.999571 | 11919 | 2 | 4 | 2065 | 0.999832 | 0.999665 | 0.999748 | 0.999744 |
| | | XGBOOST | 0.999786 | 11903 | 1 | 2 | 2084 | 0.999916 | 0.999832 | 0.999874 | 0.999962 |
| | | SVM | 0.99857 | 11885 | 7 | 13 | 2085 | 0.999411 | 0.998907 | 0.999159 | 0.998787 |
| | UDP | RF | 0.999815 | 4494 | 4 | 5 | 1997 | 0.999811 | 0.999889 | 0.999993 | 0.999972 |
| | | DT | 0.998923 | 4459 | 2 | 5 | 2034 | 0.999552 | 0.99888 | 0.999216 | 0.998947 |
| | | kNN | 0.999538 | 4435 | 2 | 1 | 2062 | 0.999549 | 0.999775 | 0.999662 | 0.999642 |
| | | XGBOOST | 0.998923 | 4400 | 3 | 4 | 2093 | 0.999319 | 0.999092 | 0.999205 | 0.999741 |
| | | SVM | 0.996769 | 4420 | 6 | 12 | 4420 | 0.99849 | 0.997548 | 0.998019 | 0.998866 |
| | HTTP GET | RF | 0.999784 | 21087 | 3 | 2 | 2008 | 0.999858 | 0.999905 | 0.999881 | 0.999862 |
| | | DT | 0.99961 | 21042 | 4 | 5 | 2049 | 0.99981 | 0.999762 | 0.999786 | 0.999649 |
| | | kNN | 0.99974 | 21025 | 1 | 5 | 2069 | 0.999952 | 0.999762 | 0.999857 | 0.999824 |
| | | XGBOOST | 0.99974 | 20983 | 1 | 5 | 2111 | 0.999952 | 0.999762 | 0.999857 | 0.999743 |
| | | SVM | 0.999351 | 20992 | 8 | 10 | 20992 | 0.998409 | 0.998182 | 0.998295 | 0.999371 |
| | DNS tunneling | RF | 0.999931 | 3187 | 3 | 1 | 2009 | 0.99976 | 0.999886 | 0.999773 | 0.999842 |
| | | DT | 0.999231 | 3170 | 2 | 2 | 2026 | 0.999369 | 0.999369 | 0.999369 | 0.999636 |
| | | kNN | 0.999231 | 3125 | 1 | 3 | 2071 | 0.99968 | 0.999041 | 0.99936 | 0.999325 |
| | | XGBOOST | 0.998654 | 3084 | 2 | 5 | 2109 | 0.999352 | 0.998381 | 0.998866 | 0.998948 |
| | | SVM | 0.995962 | 3137 | 7 | 9 | 3137 | 0.998649 | 0.995688 | 0.997166 | 0.998563 |

**Table 16.** Classification results (thermostat—Dark Nexus).

| Device/ Botnet | Attack | Algorithm | Accuracy | TP | FP | FN | TN | Precision | Recall | F1 Score | AUC |
|---|---|---|---|---|---|---|---|---|---|---|---|
| Thermostat/ Dark nexus | TCP | RF | 0.999067 | 5484 | 5 | 2 | 2009 | 0.999889 | 0.999735 | 0.999862 | 0.999807 |
| | | DT | 0.999333 | 5457 | 4 | 1 | 2038 | 0.999268 | 0.999817 | 0.999542 | 0.999866 |
| | | kNN | 0.999067 | 5440 | 2 | 5 | 2053 | 0.999632 | 0.999082 | 0.999357 | 0.999787 |
| | | XGBOOST | 0.9988 | 5409 | 4 | 5 | 2082 | 0.999261 | 0.999076 | 0.999169 | 0.999392 |
| | | SVM | 0.9972 | 5386 | 9 | 12 | 2093 | 0.998332 | 0.997777 | 0.998054 | 0.998168 |
| | UDP | RF | 0.99988 | 10188 | 5 | 5 | 2002 | 0.99988 | 0.99988 | 0.99988 | 0.99988 |
| | | DT | 0.999344 | 10146 | 3 | 5 | 2046 | 0.999704 | 0.999507 | 0.999606 | 0.999544 |
| | | kNN | 0.999262 | 10140 | 5 | 4 | 2051 | 0.999507 | 0.999606 | 0.999556 | 0.999830 |
| | | XGBOOST | 0.999344 | 10130 | 5 | 3 | 2062 | 0.999507 | 0.999704 | 0.999605 | 0.999510 |
| | | SVM | 0.998033 | 10073 | 3 | 16 | 10073 | 0.99849 | 0.996984 | 0.997736 | 0.998590 |
| | HTTP GET | RF | 0.999633 | 19765 | 3 | 5 | 2017 | 0.999848 | 0.999747 | 0.999798 | 0.999937 |
| | | DT | 0.999541 | 19730 | 5 | 5 | 2050 | 0.999747 | 0.999747 | 0.999747 | 0.999730 |
| | | kNN | 0.999725 | 19716 | 2 | 4 | 2068 | 0.999899 | 0.999797 | 0.999848 | 0.999950 |
| | | XGBOOST | 0.999679 | 19680 | 3 | 4 | 2103 | 0.999848 | 0.999797 | 0.999822 | 0.999980 |
| | | SVM | 0.999082 | 19644 | 4 | 13 | 19644 | 0.998182 | 0.997275 | 0.997728 | 0.999110 |
| | DNS tunneling | RF | 0.999649 | 9381 | 3 | 1 | 2015 | 0.99968 | 0.999893 | 0.999787 | 0.999683 |
| | | DT | 0.999737 | 9345 | 1 | 2 | 2052 | 0.999893 | 0.999786 | 0.99984 | 0.999960 |
| | | kNN | 0.999211 | 9335 | 4 | 5 | 2056 | 0.999572 | 0.999465 | 0.999518 | 0.999830 |
| | | XGBOOST | 0.999561 | 9305 | 3 | 2 | 2090 | 0.999678 | 0.999785 | 0.999731 | 0.999730 |
| | | SVM | 0.998421 | 9339 | 4 | 13 | 2044 | 0.998379 | 0.996763 | 0.99757 | 0.999860 |

**Table 17.** Classification results (camcorder—Mirai).

| Device/ Botnet | Attack | Algorithm | Accuracy | TP | FP | FN | TN | Precision | Recall | F1 Score | AUC |
|---|---|---|---|---|---|---|---|---|---|---|---|
| Camcorder/ Mirai | TCP | RF | 0.999292 | 3639 | 2 | 2 | 2007 | 0.999451 | 0.999451 | 0.999451 | 0.99907 |
| | | DT | 0.998584 | 3623 | 4 | 4 | 2019 | 0.998897 | 0.998897 | 0.998897 | 0.999863 |
| | | kNN | 0.999646 | 3598 | 1 | 1 | 2050 | 0.999722 | 0.999722 | 0.999722 | 0.999781 |
| | | XGBOOST | 0.998938 | 3592 | 3 | 3 | 2052 | 0.999166 | 0.999166 | 0.999166 | 0.999396 |
| | | SVM | 0.996106 | 3560 | 8 | 14 | 2068 | 0.997758 | 0.996083 | 0.99692 | 0.998166 |
| | UDP | RF | 0.999872 | 7545 | 2 | 4 | 1999 | 0.999835 | 0.99987 | 0.999603 | 0.99983 |
| | | DT | 0.999267 | 7497 | 5 | 2 | 2046 | 0.999334 | 0.999733 | 0.999533 | 0.999701 |
| | | kNN | 0.998953 | 7451 | 5 | 5 | 2089 | 0.999329 | 0.999329 | 0.999329 | 0.999833 |
| | | XGBOOST | 0.999581 | 7444 | 1 | 3 | 2102 | 0.999866 | 0.999597 | 0.999731 | 0.999515 |
| | | SVM | 0.997592 | 7412 | 6 | 9 | 7412 | 0.998867 | 0.996796 | 0.99783 | 0.998597 |
| | HTTP GET | RF | 0.999529 | 6479 | 2 | 2 | 2017 | 0.999691 | 0.999691 | 0.999691 | 0.99993 |
| | | DT | 0.999412 | 6461 | 2 | 3 | 2034 | 0.999691 | 0.999536 | 0.999613 | 0.999737 |
| | | kNN | 0.999412 | 6437 | 3 | 2 | 2058 | 0.999534 | 0.999689 | 0.999612 | 0.999959 |
| | | XGBOOST | 0.999412 | 6421 | 2 | 3 | 2074 | 0.999689 | 0.999533 | 0.999611 | 0.999982 |
| | | SVM | 0.998353 | 6405 | 7 | 10 | 6405 | 0.998409 | 0.998409 | 0.998409 | 0.999113 |
| | DNS tunneling | RF | 0.999249 | 5978 | 4 | 2 | 2006 | 0.999331 | 0.999666 | 0.999498 | 0.99968 |
| | | DT | 0.998999 | 5959 | 4 | 4 | 2023 | 0.999329 | 0.999329 | 0.999329 | 0.999963 |
| | | kNN | 0.999124 | 5942 | 4 | 3 | 2041 | 0.999327 | 0.999495 | 0.999411 | 0.999832 |
| | | XGBOOST | 0.998874 | 5914 | 5 | 4 | 2067 | 0.999155 | 0.999324 | 0.99924 | 0.999737 |
| | | SVM | 0.997121 | 5914 | 5 | 8 | 5914 | 0.99757 | 0.996225 | 0.996897 | 0.999861 |

**Table 18.** Classification results (camcorder—Gafgyt).

| Device/ Botnet | Attack | Algorithm | Accuracy | TP | FP | FN | TN | Precision | Recall | F1 Score | AUC |
|---|---|---|---|---|---|---|---|---|---|---|---|
| Camcorder/ Gafgyt | TCP | RF | 0.999971 | 11981 | 5 | 1 | 2003 | 0.999983 | 0.999917 | 0.99985 | 0.999889 |
| | | DT | 0.999643 | 11941 | 1 | 4 | 2044 | 0.999916 | 0.999665 | 0.999791 | 0.999486 |
| | | kNN | 0.999571 | 11924 | 5 | 1 | 2060 | 0.999581 | 0.999916 | 0.999748 | 0.999991 |
| | | XGBOOST | 0.999571 | 11916 | 3 | 3 | 2068 | 0.999748 | 0.999748 | 0.999748 | 0.999364 |
| | | SVM | 0.998713 | 11920 | 10 | 8 | 2052 | 0.999162 | 0.999329 | 0.999246 | 0.999484 |
| | UDP | RF | 0.999923 | 4496 | 4 | 3 | 1997 | 0.999811 | 0.999833 | 0.999822 | 0.999913 |
| | | DT | 0.998615 | 4465 | 5 | 4 | 2026 | 0.998881 | 0.999105 | 0.998993 | 0.999888 |
| | | kNN | 0.999077 | 4430 | 1 | 5 | 2064 | 0.999774 | 0.998873 | 0.999323 | 0.999442 |
| | | XGBOOST | 0.999231 | 4387 | 2 | 3 | 2108 | 0.999544 | 0.999317 | 0.99943 | 0.999591 |
| | | SVM | 0.997231 | 4365 | 8 | 11 | 1193 | 0.998301 | 0.998301 | 0.998301 | 0.997732 |
| | HTTP GET | RF | 0.99974 | 21056 | 3 | 3 | 2038 | 0.999858 | 0.999858 | 0.999858 | 0.999965 |
| | | DT | 0.999827 | 21049 | 1 | 3 | 2047 | 0.999952 | 0.999857 | 0.999905 | 0.999425 |
| | | kNN | 0.999784 | 21006 | 4 | 1 | 2089 | 0.99981 | 0.999952 | 0.999881 | 0.999628 |
| | | XGBOOST | 0.99974 | 20958 | 5 | 1 | 2136 | 0.999761 | 0.999952 | 0.999857 | 0.999901 |
| | | SVM | 0.999091 | 21005 | 8 | 20 | 21005 | 0.998636 | 0.996596 | 0.997615 | 0.999821 |
| | DNS tunneling | RF | 0.999038 | 3182 | 4 | 1 | 2013 | 0.999745 | 0.999686 | 0.999215 | 0.999491 |
| | | DT | 0.998077 | 3176 | 5 | 5 | 2014 | 0.998428 | 0.998428 | 0.998428 | 0.999020 |
| | | kNN | 0.998846 | 3165 | 2 | 4 | 2029 | 0.999368 | 0.998738 | 0.999053 | 0.999290 |
| | | XGBOOST | 0.998462 | 3160 | 4 | 4 | 2032 | 0.998736 | 0.998736 | 0.998736 | 0.999390 |
| | | SVM | 0.996923 | 3123 | 5 | 9 | 3123 | 0.998379 | 0.997301 | 0.99784 | 0.997460 |

**Table 19.** Classification results (camcorder—Dark Nexus).

| Device/ Botnet | Attack | Algorithm | Accuracy | TP | FP | FN | TN | Precision | Recall | F1 Score | AUC |
|---|---|---|---|---|---|---|---|---|---|---|---|
| Camcorder/ Dark Nexus | TCP | RF | 0.999722 | 5423 | 5 | 1 | 2071 | 0.999779 | 0.999816 | 0.999747 | 0.99986 |
| | | DT | 0.998933 | 5403 | 3 | 5 | 2089 | 0.999445 | 0.999075 | 0.99926 | 0.999869 |
| | | kNN | 0.9988 | 5378 | 4 | 5 | 2113 | 0.999257 | 0.999071 | 0.999164 | 0.999911 |
| | | EGB | 0.9992 | 5359 | 1 | 5 | 2135 | 0.999813 | 0.999068 | 0.999441 | 0.999932 |
| | | SVM | 0.997867 | 5336 | 10 | 6 | 2148 | 0.998129 | 0.998877 | 0.998503 | 0.999542 |
| | UDP | RF | 0.999918 | 10187 | 5 | 5 | 2003 | 0.999809 | 0.999839 | 0.999851 | 0.999861 |
| | | DT | 0.99959 | 10139 | 4 | 1 | 2056 | 0.999606 | 0.999901 | 0.999753 | 0.998747 |
| | | kNN | 0.999426 | 10132 | 4 | 3 | 2061 | 0.999605 | 0.999704 | 0.999655 | 0.999406 |
| | | EGB | 0.999262 | 10124 | 4 | 5 | 2067 | 0.999605 | 0.999506 | 0.999556 | 0.999904 |
| | | SVM | 0.998115 | 10105 | 7 | 16 | 2072 | 0.999308 | 0.998419 | 0.998863 | 0.999489 |
| | HTTP GET | RF | 0.999633 | 19769 | 3 | 5 | 2013 | 0.999848 | 0.999747 | 0.999798 | 0.999851 |
| | | DT | 0.999587 | 19733 | 5 | 4 | 2048 | 0.999747 | 0.999797 | 0.999772 | 0.999996 |
| | | kNN | 0.999862 | 19726 | 1 | 2 | 2061 | 0.999949 | 0.999899 | 0.999924 | 0.999952 |
| | | EGB | 0.999633 | 19709 | 4 | 4 | 2073 | 0.999797 | 0.999797 | 0.999797 | 0.999766 |
| | | SVM | 0.999036 | 19704 | 8 | 13 | 2065 | 0.999594 | 0.999341 | 0.999467 | 0.999123 |
| | DNS tunneling | RF | 0.999474 | 9385 | 4 | 2 | 2009 | 0.999574 | 0.999787 | 0.99968 | 0.999921 |
| | | DT | 0.999386 | 9344 | 5 | 2 | 2049 | 0.999465 | 0.999786 | 0.999626 | 0.998696 |
| | | kNN | 0.999649 | 9318 | 3 | 1 | 2078 | 0.999678 | 0.999893 | 0.999785 | 0.999282 |
| | | EGB | 0.999386 | 9305 | 2 | 5 | 2088 | 0.999785 | 0.999463 | 0.999624 | 0.999645 |
| | | SVM | 0.998421 | 9317 | 8 | 10 | 2065 | 0.999142 | 0.998928 | 0.999035 | 0.998664 |

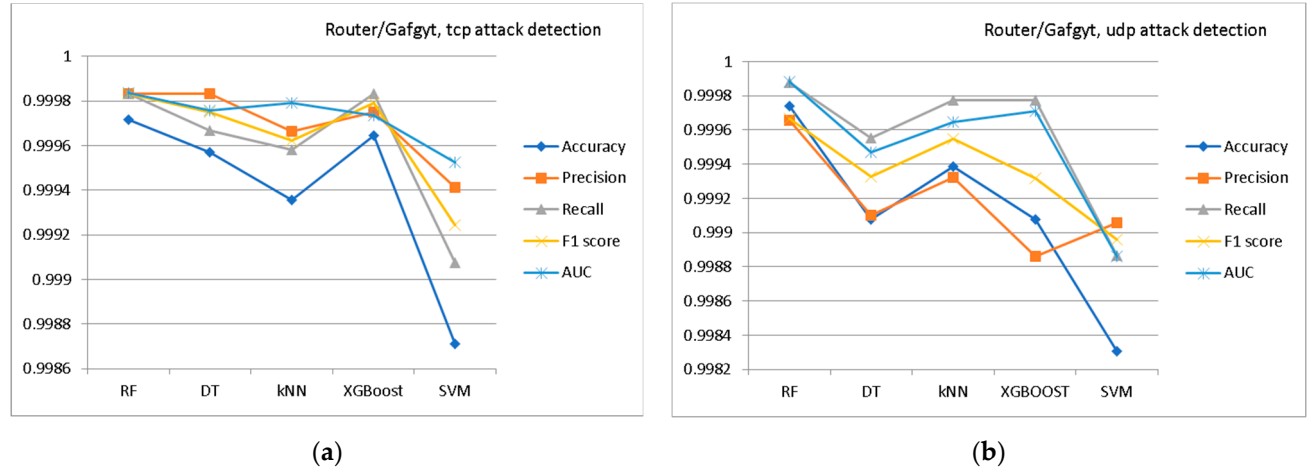

**Figure 2.** Comparison of different MLA efficiencies (decision tree—DT, random forest—RF, K-nearest neighbor—KNN, extreme gradient boosting—XGBoost, support vector machine—SVM) for Router/Mirai botnet detection: (**a**) TCP attack; (**b**) UDP attack; (**c**) HTTP GET attack; (**d**) DNS tunneling.

**Figure 3.** *Cont.*

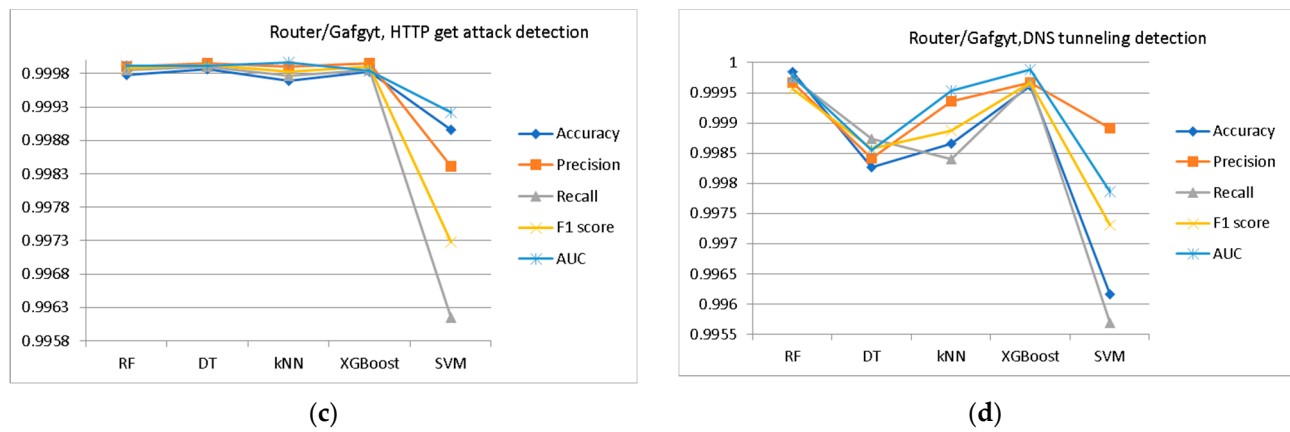

**Figure 3.** Comparison of different MLA efficiencies (decision tree—DT, random forest—RF, K-nearest neighbor—KNN, extreme gradient boosting—XGBoost, support vector machine—SVM) for Router/Gafgyt botnet detection: (**a**) TCP attack; (**b**) UDP attack; (**c**) HTTP GET attack; (**d**) DNS tunneling.

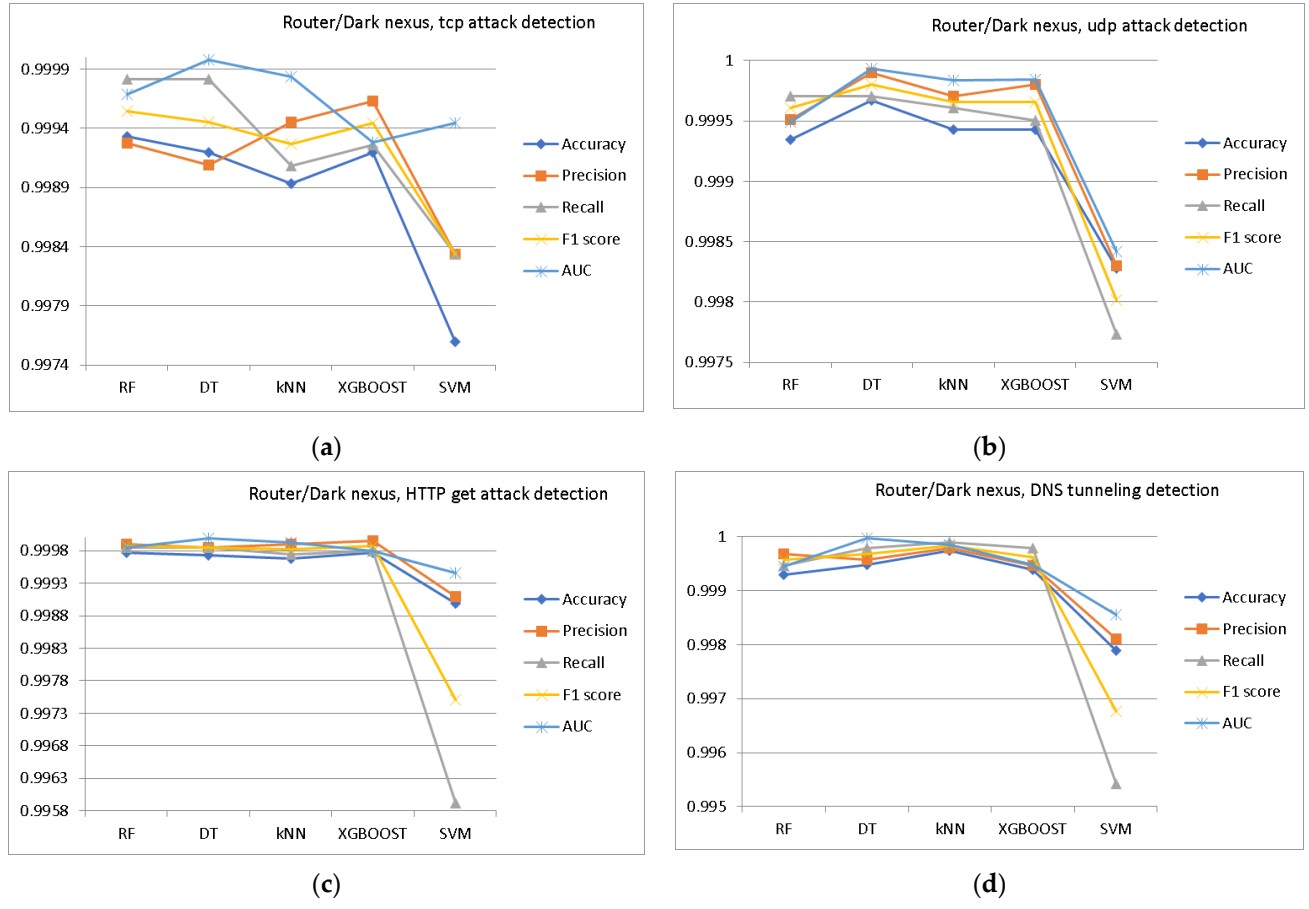

**Figure 4.** Comparison for different MLA efficiencies (decision tree—DT, random forest—RF, K-nearest neighbor—KNN, extreme gradient boosting—XGBoost, support vector machine—SVM) for Router/Dark Nexus botnet detection: (**a**) TCP attack; (**b**) UDP attack; (**c**) HTTP GET attack; (**d**) DNS tunneling.

## 6. Conclusions and Future Work

A flow-based traffic analysis allows detecting malicious behavior without the need for an in-depth packet analysis. Meanwhile, a packet content analysis provides an opportunity to decide whether the intercepted traffic belongs to the attack traffic or normal traffic in cases where the flow-based analysis does not give an unambiguous result. Attempting to

cover features (as many as possible) that indicate the presence of attacks in the Internet of Things infrastructure has its weaknesses. Such an approach requires some time to analyze in-depth, and it is poorly scalable.

The main experiment results concerning MLA involvement showed that SVM demonstrated the worst results, while the RF algorithm demonstrated the best results.

In addition, the involvement of different IoT multi-vector cyberattack features based on flow analysis and features based on the most commonly used IoT protocols caused the detection of TCP, UDP, HTTP GET, and DNS tunneling attacks approximately at the same level.

In this paper, we reviewed the known approaches to detect attacks on the Internet of Things infrastructure based on machine learning and investigated their effectiveness. We investigated the possibility of detecting traffic attacks on the Internet of Things infrastructure based on flow analysis and the most commonly used IoT protocols, such as HTTP, MQTT, and DNS.

Traffic from well-known botnets, such as Mirai, Dark Nexus, and Gafgyt was taken from well-known databases that represent common attacks on the Internet of Things infrastructures, such as TCP, UDP, HTTP GET, and DNS tunneling, used as malicious traffic.

In addition, attack traffic was generated using known utilities, and benign IoT traffic was collected from devices such as a router, a thermostat, and a camcorder.

The features presented in the work were classified using various methods of machine learning and were removed from the received traffic.

The levels of detection of the multi-vector attacks on the Internet of Things infrastructure largely depend on the involved objects of training and test samplings/settings of machine learning algorithms. This important aspect is the subject of further research.

Therefore, future work will focus on the following issues:

1. Different Internet of Things protocols [64] to remove signs of traffic, which will improve the accuracy of attack detection in the lack of flow-based analysis cases;
2. Efficient ways to reduce the number of traffic features sufficient to detect attacks;
3. Development of ML-based methods for dependability assurance of IoT systems by combining attacks and intrusion detection, redundancy, and recovery procedures [65].

**Author Contributions:** Data curation K.B. and V.K.; formal analysis S.L.; investigation K.B. and O.S.; methodology K.B. and S.L.; project administration V.K.; Software K.B.; supervision V.K.; validation K.B. and O.S.; visualization K.B. and S.L.; writing—original draft K.B. and S.L. All authors have read and agreed to the published version of the manuscript.

**Funding:** This research received no external funding.

**Institutional Review Board Statement:** Not applicable.

**Informed Consent Statement:** Informed consent was obtained from all subjects involved in the study.

**Data Availability Statement:** The dataset used for this study is publicly available at [43–49].

**Acknowledgments:** This work was supported by the ECHO project, which has received funding from the European Union's Horizon 2020 research and innovation program under the grant agreement no 830943. The authors appreciate the scientific society of the consortium for creative analysis and discussion during the preparation of this paper.

**Conflicts of Interest:** The authors declare no conflict of interest.

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
