# Peer review of "IoT Multi-Vector Cyberattack Detection Based on Machine Learning Algorithms: Traffic Features Analysis, Experiments, and Efficiency"

_algorithms, doi:10.3390/a15070239_

Round 1

Reviewer 1 Report

The manuscript “IoT Multi-Vector Cyberattacks Detection Based on Machine Learning Algorithms: Traffic Features Analysis, Experiments, and Efficiency” is interesting and well organized. However, I am writing some minor concerns and suggestions for improvement of the manuscript before final acceptance.

1.     Section 1.2: Review: Please summarize the limitations of existing studies at the end of this section.

2.     Highlight the novel contributions of the work in bullets or small paragraphs.

3.     The authors utilized five machine learning algorithms (DT, RF, KNN, XGBoost, SVM). What is the main motivation to only consider these algorithms for experiments?

4.      The authors utilized several datasets (UCI Machine Learning Repository, DS2OS, Bot-IoT, 395 N-BaIoT, CIDDS, UNSW-NB15, NSL-KDD datasets) for experiment and performance. Each dataset contains different samples and features. Please describe the preprocessing and feature selection processes for these datasets.

5.     The authors didn’t add any block diagram or flow chart to explain the proposed design. Please add and describe the attack detection process with the proposed scheme.  

6.     Please add a short description of utilized datasets along with the distribution of each dataset.

7.     Please elaborate on the values of utilized hyperparameters for optimal training and experiments of the utilized algorithms. To better understand the hyperparameters in experiments please see and cite the below paper.

·       https://ieeexplore.ieee.org/abstract/document/9626591

8.     Please add the details about the implementation platform. 

Reviewer 2 Report

In the submitted manuscript, titled "IoT Multi-Vector Cyberattacks Detection Based on Machine Learning Algorithms: Traffic Features Analysis, Experiments and Efficiency”, the authors propose an approach for IoT cyberattacks detection based on machine learning algorithms, claiming that the performed experiments demonstrate the effectiveness of their approach.

As a first observation, in order not to weigh down its exposition, I would like to suggest to the authors to avoid excessive use of acronyms in the Abstract, even more so if these have been defined by the authors and are not already known in the literature.

The proposed manuscript is not carefully written, I suggest to the authors a careful re-reading of the entire manuscript in order to fix many typos and/or grammatical forms (also in the title there is an error since they write "Traffic Features Analysis, Experiments and Efficiency" instead of "Traffic Features Analysis, Experiments, and Efficiency" ), for instance:

- “usage are cyber security risks.” instead of “usage is the cyber security risk.”;

- “a great amount of such devices” instead of “ a great number of such devices”;

- “IoT devices security” instead of “IoT device security”;

- “devices hacking” instead of “device hacking”;

- “new approaches, algorithms to assure” instead of “new approaches and algorithms to assure” or “new approaches to assure”;

- “high security level” instead of “high-security level”;

- “IoT cyberattacks detection” instead of “IoT cyberattack detection”;

- “Proposed approach deals” instead of “The proposed approach deals”;

- “presence of the cyberattacks in the IoT” instead of “presence of cyberattacks in the IoT”;

- “attack presence the set of” instead of “attack presence, a set of”;

- “were employed.” instead of “was employed.”;

- “Series of experiments” instead of “A series of experiments”;

- “NFF based approach” instead of “NFF-based approach”;

- ... And many others, considering that I recognized the above problems only in the Abstract: for this reason, I suggest involving a native-English speaker in the above process, as the manuscript presents many English errors and not simple typos, e.g., the title of the subsection 1.2 is incorrect and it must be “The State of the Art” since “state of the art” is a noun phrase. In English, nouns can become adjectives through the process of hyphenation, then the hyphenated state-of-the-art is an adjective phrase.

To improve the manuscript readability, I suggest reorganizing the "Introduction" section into only two sections, "Introduction" and "Background and Related Work", devoting the “Introduction” only for a brief overview of the domain taken into account, expanding each concept using subsections of the "Background and Related Work" one.

More generally, I would avoid using subsections where they are unnecessary, as it weighs down the reading of the manuscript: for instance, the two subsections in the "Conclusions", a section that could be simply renamed "Conclusions and Future Work".

In addition, the authors should cite and discuss further literature works that are focused on information very close or directly related to the research area taken into account, as to offer an overview to the readers.

The references appear quite updated but according to my previous observation, the authors should cite and discuss additional literature works that are close or directly related to the considered domain, such as, just by way of example:

(-) Liu, Hongyu, and Bo Lang. "Machine learning and deep learning methods for intrusion detection systems: A survey." applied sciences 9.20 (2019): 4396.

(-) Saia, Roberto, et al. "A Probabilistic-driven Ensemble Approach to Perform Event Classification in Intrusion Detection System." KDIR. 2018.

(-) Abdulhammed, Razan, et al. "Features dimensionality reduction approaches for machine learning based network intrusion detection." Electronics 8.3 (2019): 322.

(-) Abdulhammed, Razan, et al. "Deep and machine learning approaches for anomaly-based intrusion detection of imbalanced network traffic." IEEE sensors letters 3.1 (2018): 1-4.

(-) Carta, Salvatore, et al. "A local feature engineering strategy to improve network anomaly detection." Future Internet 12.10 (2020): 177.

(-) … And so on.

In conclusion, the proposed work presents weaknesses both from a formal and a substantial point of view:

(i) as for the formal aspect, it is necessary that the authors involve a native English speaker for an in-deep revision of the entire manuscript with regard to the English aspect, as well as to rewrite some sentences using more appropriate terminology.

(ii) as regards the substantive aspect, the description of the validation process of the proposed approach appears unclear but, more generally, all the contents are presented in an unclear and sometimes confusing way; there is excessive use of irrelevant information, as well as many repetitions of terms; the validation process uses unsuitable evaluation metrics for the context taken into account, such as, for instance, the Accuracy, which is a metric that does not provide a reliable performance assesment in case of unbalanced data, whereas more suitable metrics such as AUC are not taken into account by the authors.

For the above main reasons (other minor problems affect the manuscript), I can not express a positive opinion, also because the scientific contribution related to the proposed work is not clear and must be clear defined by the authors with regard to both the current literature and the possible real-world applications.

Round 2

Reviewer 2 Report

Although the revision made by the authors has solved some of the problems I reported in my previous review, the manuscript still has numerous grammar errors (incorrect use of the singular/plural form, incorrect use of the conjunctions and/or, incorrect use of the determinative/indefinite articles, and so on).

In addition, concerning the substantive aspect: there continues to be numerous unuseful information (e.g., the pseudocode formalization of the Random Forests algorithm) that can be avoided since they are well-known elements; the manuscript exposition continues to be quite confusing and unclear, so the authors should reorganize it in a better way, also by recurring to several graphs that make the information in the numerous tables more immediate.

In summary, the most important problems previously reported by me have been solved in the new version of the manuscript, so I invite the authors to fix the above ones using for the formal part both a native-English speaker (as previously suggested) and a good spell/grammar checker.
